# Triggering MSR1 promotes JNK-mediated inflammation in IL-4-activated macrophages

Manman Guo[1],[†],[‡] (ID), Anetta Härtlova[1],[2],[3],[4],[†],[*] (ID), Marek Gierliński[5], Alan Prescott[6], Josep Castellvi[7], Javier Hernandez Losa[7],[8], Sine K Petersen[3],[4], Ulf A Wenzel[3],[4], Brian D Dill[1], Christoph H Emmerich[1], Santiago Ramon Y Cajal[7],[8], David G Russell[9] & Matthias Trost[1],[2],[**] (ID)

## Abstract

Alternatively activated M2 macrophages play an important role in maintenance of tissue homeostasis by scavenging dead cells, cell debris and lipoprotein aggregates via phagocytosis. Using proteomics, we investigated how alternative activation, driven by IL-4, modulated the phagosomal proteome to control macrophage function. Our data indicate that alternative activation enhances homeostatic functions such as proteolysis, lipolysis and nutrient transport. Intriguingly, we identified the enhanced recruitment of the TAK1/MKK7/JNK signalling complex to phagosomes of IL-4-activated macrophages. The recruitment of this signalling complex was mediated through K63 polyubiquitylation of the macrophage scavenger receptor 1 (MSR1). Triggering of MSR1 in IL-4-activated macrophages leads to enhanced JNK activation, thereby promoting a phenotypic switch from an anti-inflammatory to a pro-inflammatory state, which was abolished upon MSR1 deletion or JNK inhibition. Moreover, MSR1 K63 polyubiquitylation correlated with the activation of JNK signalling in ovarian cancer tissue from human patients, suggesting that it may be relevant for macrophage phenotypic shift *in vivo*. Altogether, we identified that MSR1 signals through JNK via K63 polyubiquitylation and provides evidence for the receptor's involvement in macrophage polarization.

**Keywords** macrophage scavenger receptor 1; phagosome; proteomics; scavenger receptor; tumour-associated macrophages

**Subject Categories** Post-translational Modifications, Proteolysis & Proteomics; Signal Transduction; Immunology

**The EMBO Journal (2019) 38: e100299**

## Introduction

Phagocytosis is a highly conserved process essential for host defence and tissue remodelling. It involves the recognition of particles by a variety of cell surface receptors, followed by cargo processing and delivery to lysosomes via phagosome–lysosome fusion, process known as phagosome maturation. This leads to gradual acidification of the phagosomal lumen and acquisition of digestive enzymes required for the degradation of phagosomal cargo. Therefore, phagocytosis is not only responsible for elimination of bacterial pathogens, but also responsible for the clearance of apoptotic cells, cell debris and senescence cells and orchestrates the subsequent immune response (Rothlin *et al*, 2007; Murray & Wynn, 2011; Lemke, 2013, 2017). Central to this process is phagosome function. If uncontrolled, the inappropriate clearance of apoptotic bodies can give rise to autoimmune diseases, atherosclerosis and cancer, while failure to ingest or kill pathogens can result in deadly infections (Johnson & Newby, 2009; Nagata *et al*, 2010; Colegio *et al*, 2014; Arandjelovic & Ravichandran, 2015). Therefore, it is of great importance to understand which signalling pathways regulate phagocytosis and phagosomal maturation.

It has recently been acknowledged that the phagosome serves as a signalling platform and interacts with innate immune signalling (Stuart *et al*, 2007; Martinez *et al*, 2011, 2015; Kagan, 2012; Heckmann *et al*, 2017). However, whether phagosome-associated cell signalling is independent of its role in cargo degradation has not been well understood. Supporting this notion, recent proteomic studies demonstrated that phagosomes are dynamic organelles that change their composition and function in response to infection or inflammation (Trost *et al*, 2009; Boulais *et al*, 2010; Dill *et al*, 2015; Guo *et al*, 2015; Naujoks *et al*, 2016; Hartlova *et al*, 2018). While

1  MRC Protein Phosphorylation and Ubiquitylation Unit, University of Dundee, Dundee, UK
2  Institute for Cell and Molecular Biosciences, Newcastle University, Newcastle upon Tyne, UK
3  Wallenberg Centre for Molecular and Translational Medicine, University of Gothenburg, Gothenburg, Sweden
4  Department of Microbiology and Immunology, Institute for Biomedicine, Sahlgrenska Academy, University of Gothenburg, Gothenburg, Sweden
5  Data Analysis Group, School of Life Sciences, University of Dundee, Dundee, UK
6  Division of Cell Signalling and Immunology, School of Life Sciences, University of Dundee, Dundee, UK
7  Department of Pathology, Hospital Universitario Vall d'Hebron, Barcelona, Spain
8  Spanish Biomedical Research Network Centre in Oncology (CIBERONC), Barcelona, Spain
9  Department of Microbiology and Immunology, College of Veterinary Medicine, Cornell University, Ithaca, NY, USA
  *Corresponding author. Tel: +46 31 786 6241; E-mail: anetta.hartlova@gu.se
  **Corresponding author. Tel: +44 191 2087009; E-mail: matthias.trost@ncl.ac.uk
  †These authors contributed equally to this work
  ‡Present address: Botnar Research Centre, Nuffield Department of Orthopaedics, Rheumatology and Musculoskeletal Sciences, University of Oxford, Oxford, UK

the regulation of phagosomal maturation in so-called M1 inflammatory macrophages has been extensively studied, the mechanisms facilitating phagosomal maturation in macrophages involved in tissue repair remain poorly understood (Balce *et al*, 2011).

Th2-derived cytokines, such as interleukin-4 (IL-4) and interleukin-13 (IL-13), induce a strong anti-inflammatory macrophage phenotype, also called alternative-activated macrophages (M2). M2 macrophages and tissue-resident macrophages, which often resemble an M2-like state, clear cell debris and dead cells through phagocytosis. They are therefore essential for maintenance and tissue homeostasis. M2 alternatively activated macrophages (AAMs) inhibit inflammatory responses and promote angiogenesis and tissue repair by synthetizing mediators required for collagen deposition, which is important for wound healing (Gordon & Martinez, 2010). It has been shown that IL-4 enhanced phagosomal protein degradation (Balce *et al*, 2011). Whether IL-4 regulates other phagosomal functions, and through which molecular mechanisms, remains unclear.

Here, we investigated the phagosome proteome of IL-4-activated macrophages. In line with the known role in homeostasis, the phagosome of AAMs has increased abilities to degrade incoming apoptotic cells and transport the resulting nutrients. Furthermore, we demonstrate that the TAK1/MKK7/JNK signalling complex showed an enhanced association with the phagosome upon IL-4 macrophage activation. The assembly of the signalling complex is mediated through K63 polyubiquitylation. By combining K63-polyubiquitylation enrichment and mass spectrometry approaches, we identified macrophage scavenger receptor 1 (MSR1) as the upstream receptor that promotes the recruitment of the TAK1/MKK7/JNK signalling complex to the phagosome. Triggering MSR1 induces JNK activation in M2 macrophages. This MSR1/JNK signalling pathway activation leads to a M2/M1 macrophage phenotypic switch that is abolished in macrophages lacking MSR1. We demonstrate that MSR1 is K63-ubiquitylated and signals through JNK in human patient ovarian cancer, thus suggesting a potential role in human cancer.

# Results

## Alternative activation regulates phagosomal proteolysis and lipolysis

To determine the impact of IL-4 on phagocytosis and phagosomal functions, we examined the rate of phagocytosis and phagosomal functions in IL-4 AAMs (M2) and resting macrophages (M0). We found that both IL-4- and IL-13-activated M2 macrophages have enhanced uptake of apoptotic cells, while uptake of necrotic cells was comparable to M0 resting macrophages (Fig 1A). To determine whether the enhanced uptake was because of the negative charge of apoptotic cells, we compared the uptake of fluorescently labelled carboxylated negatively charged and positively charged amino silica microspheres in M2 and M0 macrophages. The analysis revealed an increased uptake of negatively charged microspheres in M2 macrophages, which are taken up through scavenger receptors (Tanaka *et al*, 1996; Platt *et al*, 1999; Stephen *et al*, 2010), while the engulfment of positively charged microspheres was similar to M0 macrophages (Fig 1B). This indicates that, due to their similar

uptake behaviour, carboxylated microspheres may serve as a surrogate for apoptotic cells (Kiss *et al*, 2006). Next, we analysed the functional parameters of the phagosomal lumen. In these assays, we use fluorescent probes that allow the measurement of proteolysis (a readout for maturation), acidification and lipolysis in real time (Yates *et al*, 2005; Podinovskaia *et al*, 2013). Consistent with the previous reports, we observed enhanced proteolytic activity in phagosomes of M2 macrophages (Balce *et al*, 2011). Furthermore, we found that IL-4 increased phagosomal lipid degradation and facilitated phagosomal acidification (Fig 1C–E) indicating that IL-4 activation promotes phagosome maturation and the ability of macrophages to degrade lipid-rich particles through phagosomes. Altogether, these data indicate that that alternative activation increases the ability to take up and degrade apoptotic cells and other lipid-rich particles by increasing the degradative potential of phagolysosomes from M2 (IL4) MΦs.

## Quantitative proteomic analysis of phagosomes from IL-4 alternatively activated macrophages

To obtain further molecular details about the changes on phagosomes of M2 macrophages, we isolated highly pure phagosomes from M2 and M0 macrophages by floatation on a sucrose gradient using carboxylated microspheres and analysed their proteomes by quantitative LC-MS/MS (Fig 2A; Appendix Figs S1A–C and S2A; Desjardins *et al*, 1994; Peltier *et al*, 2017; Trost *et al*, 2009). Comparative analysis led to the identification of 20,614 distinct peptides corresponding to 1,948 unique proteins across three independent replicates at a false-discovery rate (FDR) of < 1%, of which 1,766 proteins were quantified in at least two of the three biological replicates. IL-4 activation induced strong changes to the phagosome proteome with 121 proteins significantly up-regulated and 62 proteins significantly down-regulated (twofold change, $P < 0.05$; Fig 2B; Dataset EV1), some of which we validated by Western blot analysis (Appendix Fig S2B). Consistent with the above observations, a subset of proteins involved in lipid metabolism (Lpl lipoprotein lipase, ABHD12 lipase and phospholipase D1), acidification (v-ATPase complex) and lysosomal enzymes including cathepsins L1 and D were highly enriched on the phagosome of M2 macrophages (Fig 2C; Dataset EV1). Moreover, GO term (Fig 2D) and protein network analysis (Appendix Fig S2C) further showed that IL-4 alternative activation also increased phagosome abundance of scavenger receptors such as MARCO, CD36, Colec12 and MSR1 required for clearance of dead cells while Toll-like receptors (TLRs) involved in inflammatory response were reduced (Fig 2D, Appendix Fig S2C). Furthermore, M2 phagosomes showed higher phosphatidylinositol-binding proteins, suggesting changes to the phagosome membrane lipid content (Fig 2D). Consistent with a previous report, superoxide anion generation including the NADPH oxidase complex proteins NCF1 (p47-phox), Cyba (p22-phox), Cybb (gp91-phox) and superoxide dismutase SOD1 was strongly down-regulated in phagosomes from M2 macrophages (Fig 2D, Appendix Fig S2C; Balce *et al*, 2011). Interestingly, M2 macrophages also enriched a large number of specific carbohydrate-binding proteins such as lectins, while carbohydrate hydrolases were significantly reduced. This indicates a conservation of phagocytosed glycans, potentially for antigen presentation via MHC class II molecules (Avci *et al*, 2013). Taken together,

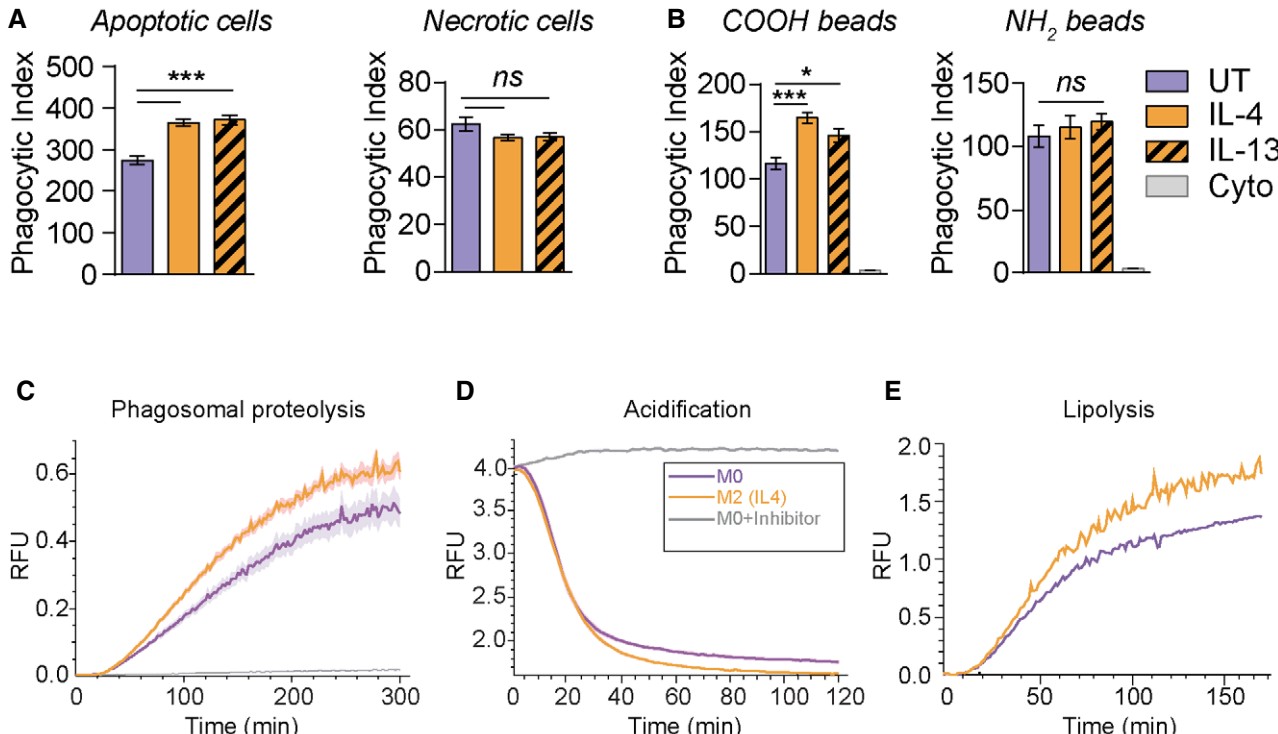

**Figure 1. Alternative activation affects phagosomal proteolysis and lipolysis as well as phagocytosis of negatively charged particle.**

A  Phagocytosis assay of apoptotic or necrotic GFP-expressing RAW264.7 cells in M2 (IL-4 or IL13) and untreated M0 BMDMs.

B  Phagocytosis of fluorescent negatively charged carboxylated and positively charged amino microspheres (B) in primary M2 (IL-4) and M0 macrophages. Cytochalasin D (Cyto) (6 μM) was used as an inhibitor of phagocytosis, 1 h before phagocytosis.

C–E  Real-time fluorescence assays for intraphagosomal proteolysis (C), acidification (D) and lipolysis (E) show substantially increased proteolysis, acidification and lipolysis in the phagosomes of M2 (IL-4) macrophages. The kinetics of proteolysis, acidification and lipolysis of phagocytosed beads were plotted as a ratio of substrate fluorescence to calibration fluorescence. Beads were added to macrophages at 0 min. (E) is a representative of three independent experiments. Leupeptin (100 nM) and bafilomycin (100 nM) treatments serve as negative controls in (C) and (D), respectively.

Data information: The statistical significance of data is denoted on graphs by asterisks where $*P < 0.05$, $***P < 0.001$ or ns = not significant. (A, B) Data are shown as means of relative fluorescence units (RFU) ± standard error of the mean (SEM), Student's *t*-test used. (C, D) Shaded area represents SEM. (A–D) Three replicates were used.

these results indicate that the phagosome of M2 macrophages has mainly a homeostatic role with its increased ability to hydrolyse proteins and lipids of incoming cargo.

### TAK1/MKK7/JNK is recruited to the phagosome of M2 macrophages via K63 polyubiquitylation

Interestingly, anti-inflammatory IL-4 activation also led to an increased phagosome abundance of the pro-inflammatory MAP kinase signalling complex around TAK1 (Map3k7, 2.1-fold) and MKK7 (Map2k7, 3.1-fold; Dataset EV1; Appendix Fig S2C) indicating cross-regulation between anti- and pro-inflammatory pathways. Given the increased abundance of these pro-inflammatory kinases was surprising on phagosomes of IL-4-stimulated macrophages, we next investigated how this complex was translocated to the phagosome. Immunoblot analyses of total cell lysates and phagosomal fractions revealed significant enrichment of TAK1 and MKK7 on phagosomes of M2 macrophages compared to resting M0 macrophages (Fig 3A and B). Activated TAK1 can phosphorylate two MAPK kinases, MKK4 and MKK7, which both can activate JNK. While MKK4 can activate p38 and JNK MAPK signalling pathways, MKK7 selectively activates JNK (Tournier *et al*, 2001). Noteworthy,

our mass spectrometry data revealed that only MKK7 was enriched on phagosomes upon IL-4 alternative activation. Consistent with our LC-MS/MS data, MKK4 was not detected on phagosomes of M2 macrophages by immunoblot analysis indicating that MKK7 alone was important in this phagosome signalling pathway. Further immunoblot analysis also confirmed enrichment of JNK of M2 macrophage phagosomes (Appendix Fig S4C).

Previous data have shown that upon pro-inflammatory interleukin-1 receptor or Toll-like receptor (TLR) activation, the TAK1/MKK7/JNK complex binds to the TAB1/TAB2 protein complex, which in turn is recruited to K63-polyubiquitin chains (Xia *et al*, 2009; Emmerich *et al*, 2013). We next tested whether TAB1/TAB2 is recruited to the phagosome of M2 macrophages. Indeed, immunoblot analysis demonstrated enrichment of TAB1/TAB2 on phagosomes of M2 macrophages compared to M0. To further validate the recruitment of the protein complex to phagosomes, we performed confocal fluorescence microscopy and showed vesicular distribution in the cytoplasm with enhanced recruitment of TAB1, TAB2 and MKK7 to the M2 macrophage phagosome compared to M0 macrophages (Appendix Fig S3A and B).

As it is well-established that TAK1 binds via TAB1/2/3 to free and protein-anchored K63-polyubiquitin chains in inflammatory

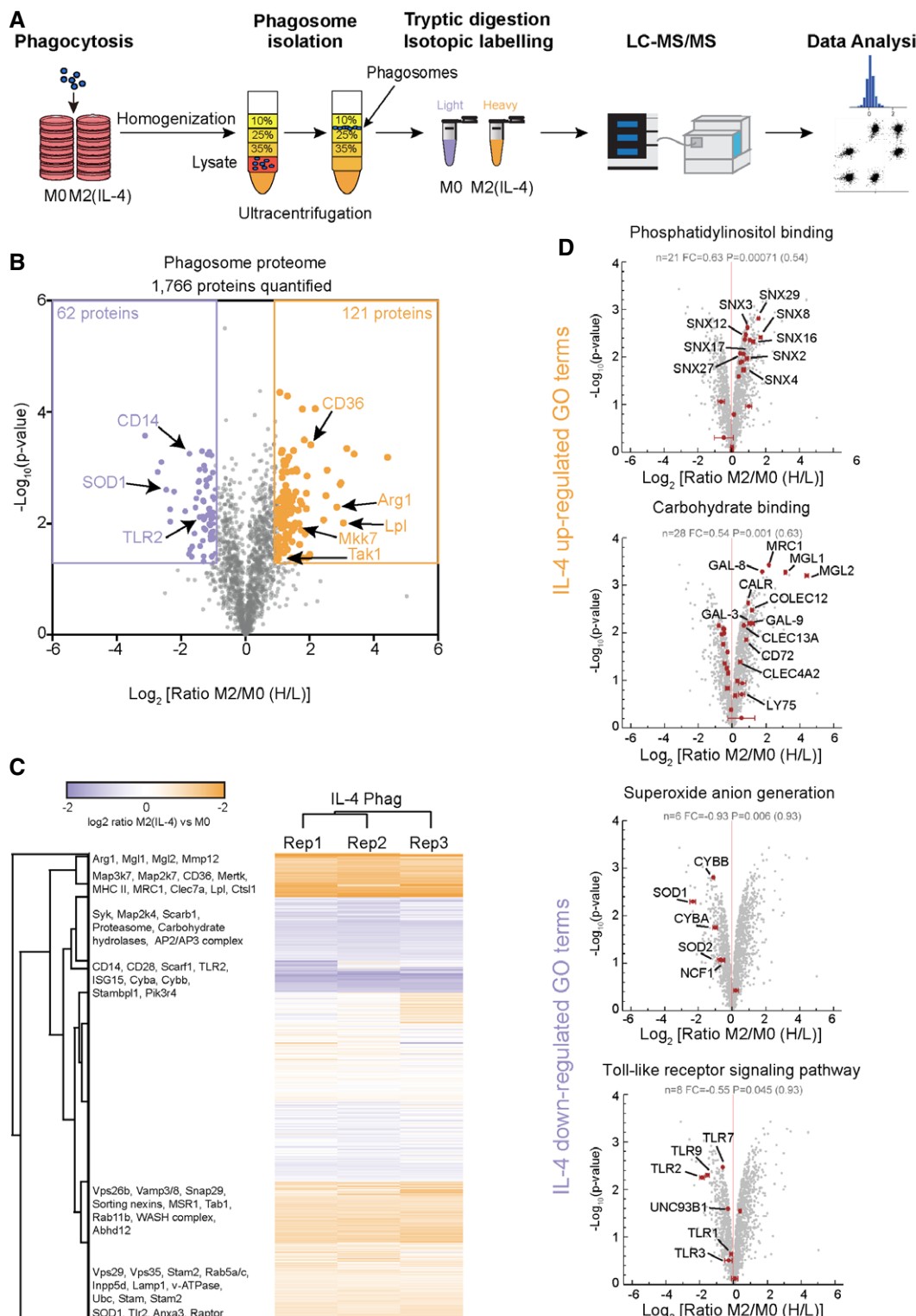

**Figure 2. Experimental workflow and the phagosome proteome of M2 (IL-4) macrophages.**

A  Workflow of the phagosome proteomic experiment.

B  Volcano plot of the phagosome proteome data. 1,766 proteins were quantified of which 121 proteins were significantly up-regulated and 62 proteins were down-regulated in M2 (IL-4) macrophages. Selected proteins are indicated.

C  Heatmap of proteomic data shows high reproducibility between biological replicates. Selected proteins are highlighted.

D  Selected Gene Ontology (GO) terms of biological processes significantly up-regulated and down-regulated on phagosomes of M2 (IL-4) macrophages. Selected proteins of these GO-terms are highlighted. Error bars represent standard deviations from three biological replicates.

innate immune responses (Xia *et al*, 2009; Emmerich *et al*, 2013), we tested whether phagosomes from M2 macrophages contain K63-polyubiquitylated proteins independent of inflammatory stimuli. Immunoblot analysis of phagosome extracts probed with anti-K63-polyubiquitin antibodies revealed that phagosomes contain a large amount of K63-polyubiquitylated proteins compared to the total cell lysate, which was even more increased by alternative activation (Fig 3C). To determine whether recruitment of TAB1, TAB2, TAK1 and MKK7 to the M2 phagosome was indeed K63-polyubiquitylation-dependent, we treated cells with

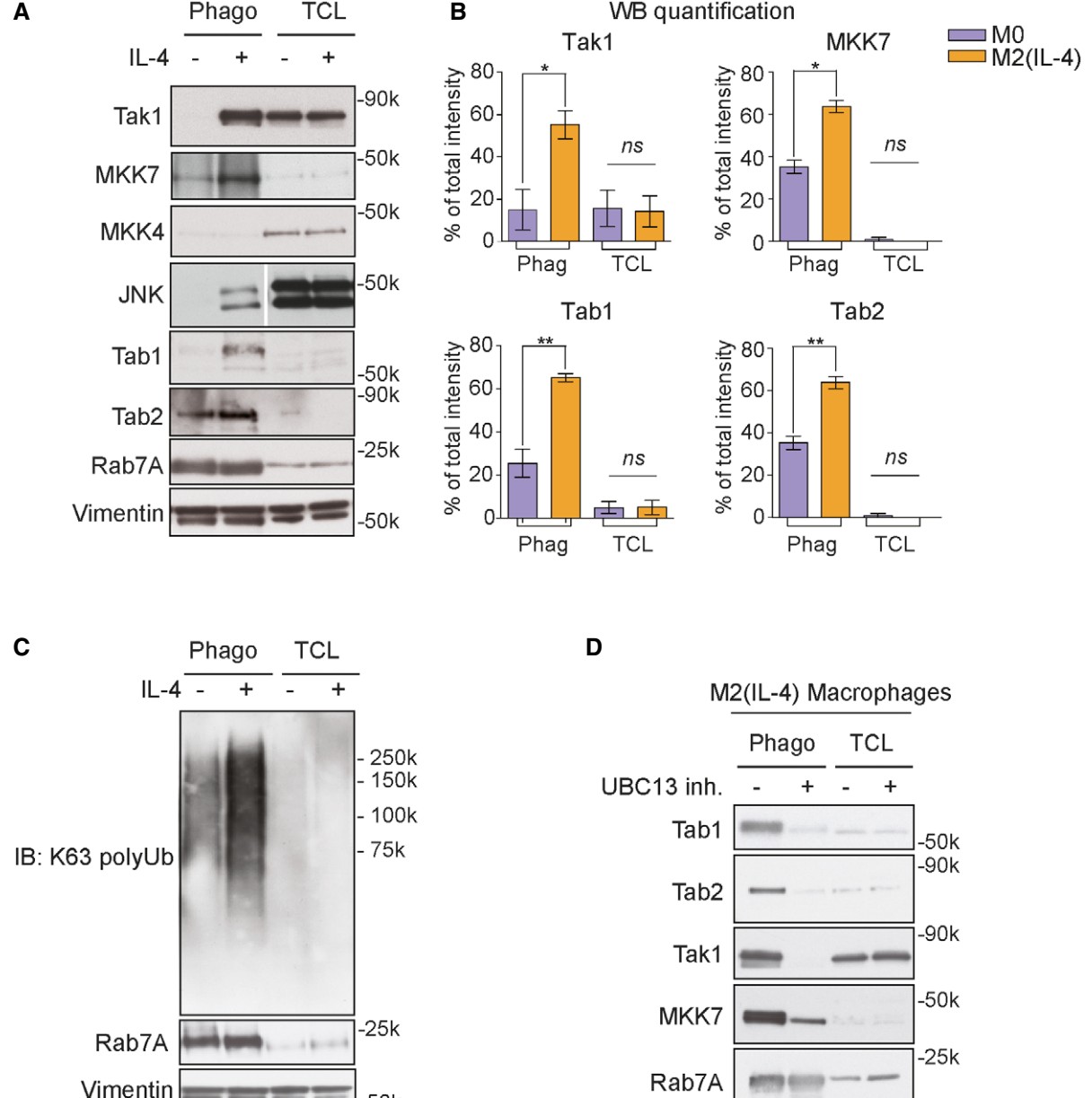

**Figure 3. TAK1/MKK7/JNK complex is recruited to phagosomes of M2 (IL-4) macrophages in a K63-polyubiquitylation-dependent manner.**

A  Immunoblot (IB) analysis showing the recruitment of TAK1, MKK7, JNK and TAB1/TAB2 to the phagosome in M2 (IL-4) macrophages, while MKK4 is not recruited. Rab7a, a phagosomal marker, was used as a loading control for phagosomes, and vimentin was used as a loading control for total cell lysates. The white line in the JNK blot shows other bands were cut out. Full blot can be seen in Appendix Figure S4.

B  Quantitation of three independent IB experiments by ImageJ of non-saturated blots for TAK1, MKK7, TAB1 and TAB2 expression on phagosomes of resting M0 macrophages and M2 (IL-4) macrophages. Error bars represent SEM. *$P < 0.01$, **$P < 0.001$ (Student's *t*-test). Three replicates were used.

C  IB showing enrichment of K63-polyubiquitylated proteins on the phagosome of M2 (IL-4) macrophages compared to M0 macrophages.

D  Treatment with the UBC13 inhibitor NSC697923 reduces recruitment of TAB1, TAB2, TAK1 and MKK7 to the phagosome of M2 (IL-4) macrophages, indicating a K63-polyubiquitylation-dependent translocation for these proteins.

Data information: (C) and (D) are representatives of three and two independent experiments, respectively.

NSC697923, a pharmacological inhibitor of the K63-specific E2-conjugating enzyme UBC13-UEV1A (Pulvino *et al*, 2012) and probed isolated phagosomes for K63 polyubiquitylation. As shown in Fig 3D, recruitment of the protein complex was virtually abolished under these conditions, which we also confirmed for MKK7 by immunofluorescence microscopy (Appendix Fig S4A and B). These data indicate that IL-4 activation of macrophages promotes K63 polyubiquitylation, which recruits the TAK1/MKK7/JNK complex to the phagosome.

### Macrophage scavenger receptor 1 is K63-polyubiquitylated and interacts with TAK1/MKK7/JNK on the phagosome of M2 macrophages

In order to identify the K63-polyubiquitylated proteins, which bind the TAK1/MKK7/JNK complex on the phagosome of M2 macrophages, we enriched polyubiquitylated phagosomal proteins from M2 macrophages using tandem ubiquitin-binding entities (TUBEs) of a repeat of the Npl4 Zinc Finger (NZF) domain of TAB2 tagged with Halo (termed here Halo-TAB2). These constructs have been shown to bind specifically to K63-polyubiquitin chains (Fig 4A; Hjerpe *et al*, 2009; Emmerich *et al*, 2013; Heap *et al*, 2017). Quantitative mass spectrometric analysis of these pull-downs identified 538 phagosomal proteins that were reproducibly captured by Halo-TAB2 compared to mutant, ubiquitin-non-binding Halo (T674A/F675A) TAB2 control beads (based on a twofold, $P < 0.05$ cut-off; Dataset EV2). Moreover, we identified 62 novel direct ubiquitylation (−GlyGly) sites on 33 different phagosomal proteins. Quantitation of the data revealed that the Gly-Gly peptide derived from K63-linked polyubiquitin was by far the most abundant, proving that we achieved good enrichment. However, we also identified peptides for K11-, K48-, linear/M1-, K29-, K6-, K27- and K33 (in order of decreasing abundances)-linked polyubiquitin, suggesting that there are either mixed chains or K63-polyubiquitylated proteins might also be modified with other polyubiquitin chains.

Other identified ubiquitylated proteins included many known phagolysosomal proteins such as the large neutral amino acid transporter SLC43A2/LAT4 (K283, K293, K402, K557), the cholesterol transporter ABCG1 (K55), Fc- and B-cell receptor adaptor LAT2 (K39, K84), LYN kinase (K20) and the TLR chaperone UNC93B1 (K197, K582; Fig 4B; Appendix Table S1).

Interestingly, one of the most abundant Gly-Gly-modified peptides was a peptide containing lysine 27 (K27) of macrophage scavenger receptor 1/scavenger receptor A (MSR1/SR-A; CD204). This site is highly conserved between human and mouse (Fig 4C). MSR1 is a multifunctional phagocytic receptor, highly expressed in macrophages, involved in uptake of apoptotic cells and modified lipoproteins (Kelley *et al*, 2014). In addition to its scavenging function, MSR1 has been implicated in the innate immune response to bacteria (Platt & Gordon, 2001).

Our MS and immunoblot data showed an increase in MSR1 on phagosomes from M2 macrophages compared to M0 macrophages (Dataset EV1; Appendix Fig S5A and C), while both total cell and cell surface expression levels of MSR1 were unchanged between the two conditions (Appendix Fig S5B). However, when we precipitated K63-polyubiquitylated proteins from resting and M2 macrophages, we could see that the polyubiquitylated forms of MSR1 were considerably more abundant in alternatively activated macrophages (Fig 4D), suggesting that MSR1 becomes more polyubiquitylated in M2 macrophages.

To validate K63 polyubiquitylation of MSR1, we treated enriched polyubiquitylated phagosome protein extracts from M2 macrophages with the K63-specific deubiquitylase (DUB) AMSH-LP or the non-specific DUB USP2 (Ritorto *et al*, 2014; Fig 4E). In both cases, the high molecular smear of ubiquitylated MSR1 decreased while the band representing the non-ubiquitylated form of MSR1 increased significantly, indicating that MSR1 was predominantly K63-polyubiquitylated on phagosomes upon uptake of carboxylated beads in M2 macrophages.

We next investigated whether K63-polyubiquitylated MSR1 might recruit the TAB1/TAB2/TAK1/MKK7 complex to the phagosome. To test this, we pulled down MSR1 from extracts of carboxylated bead phagosomes. We found that indeed TAB1/TAB2/TAK1/MKK7 complex co-immunoprecipitated with MSR1 (Fig 4F). Moreover, using a different antibody against TAK1, immunoblot analysis patterns indicated also substantial post-translational modification—most likely polyubiquitylation—of TAK1, which was considerably enhanced in MSR1 IPs. It has been reported that ubiquitylation of TAK1 activated the kinase activity (Fan *et al*, 2010), suggesting that TAK1 and the downstream kinase MKK7 are recruited in the active state or activation is triggered by binding to K63-polyubiquitylated MSR1. Taken together, these data demonstrate that phagosomal MSR1 becomes K63-polyubiquitylated upon activation in IL-4-activated macrophages, which recruits the TAK1/MKK7 kinase signalling complex.

### Triggering MSR1 activates JNK pathway

To further test whether the engagement of MSR1 causes the activation of the TAK1/MKK7/JNK pathway, wild-type (WT) and MSR1 KO M2 macrophages were stimulated with the MSR1 ligands fucoidan and oxidized LDL (oxLDL) and analysed for MSR1 K63 polyubiquitylation and activation of JNK signalling (Greaves & Gordon, 2009). Pull-downs of ubiquitylated proteins were blotted for MSR1, which revealed that MSR1 activation increased the amount its polyubiquitylation. Moreover, M2 macrophages deficient in MSR1 showed decreased JNK phosphorylation upon fucoidan or oxLDL stimulation (Fig 5A and B), indicating that MSR1 directly signals through the JNK signalling pathway. Furthermore, JNK activation was dependent on MSR1 polyubiquitylation, as K27R mutation of MSR1 abolished JNK activation in response to fucoidan in IL-4-activated macrophages (Fig 5C). This prompted us to test whether MSR1 signalling through the JNK kinase complex plays a role in affecting the inflammatory state of macrophages by characterizing gene expression and cell surface markers. Ablation of MSR1 resulted in diminished induction of the pro-inflammatory genes *Il1b, Tnfa* and *Ccl2* in response to fucoidan in M2 macrophages (Fig 5D). Consistently, MSR1 KO M2 macrophages exhibited reduced cell surface expression of pro-inflammatory markers CD69, CD86 and CD54 while the M2-cell surface markers CD36 and CD301b (Mgl2) remained unperturbed or even reduced upon MSR1 ligation (Fig 5F; Appendix Figs S6A and S7). This increase in pro-inflammatory state was abolished by treatment with the specific JNK inhibitor JNK-IN8 (Zhang *et al*, 2012; Fig 5E, Appendix Fig

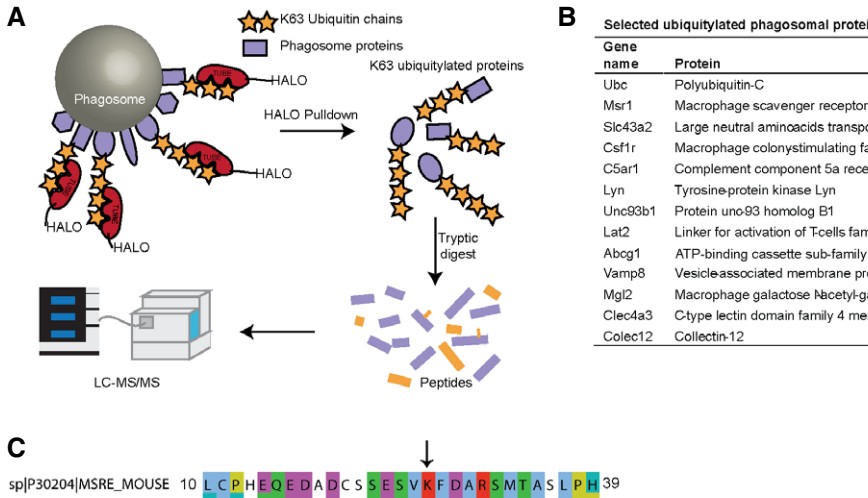

**B**

Selected ubiquitylated phagosomal proteins identified by TUBE-MS approach

| Gene name | Protein | Ubiquitylated position |
|---|---|---|
| Ubc | Polyubiquitin-C | 63;11;48;1;29;6;27;33 |
| Msr1 | Macrophage scavenger receptor 1 | 27 |
| Slc43a2 | Large neutral aminoacids transporter small subunit 4 | 283;293;402;557 |
| Csf1r | Macrophage colonystimulating factor 1 receptor | 572;810;625;584 |
| C5ar1 | Complement component 5a receptor 1 | 334 |
| Lyn | Tyrosine-protein kinase Lyn | 20 |
| Unc93b1 | Protein unc-93 homolog B1 | 197;582 |
| Lat2 | Linker for activation of T-cells family member 2 | 39;84 |
| Abcg1 | ATP-binding cassette sub-family G member 1 | 55 |
| Vamp8 | Vesicle-associated membrane protein 8 | 47 |
| Mgl2 | Macrophage galactose N-acetyl-galactosamine | 16 |
| Clec4a3 | C-type lectin domain family 4 member a3 | 13 |
| Colec12 | Collectin-12 | 2;17 |

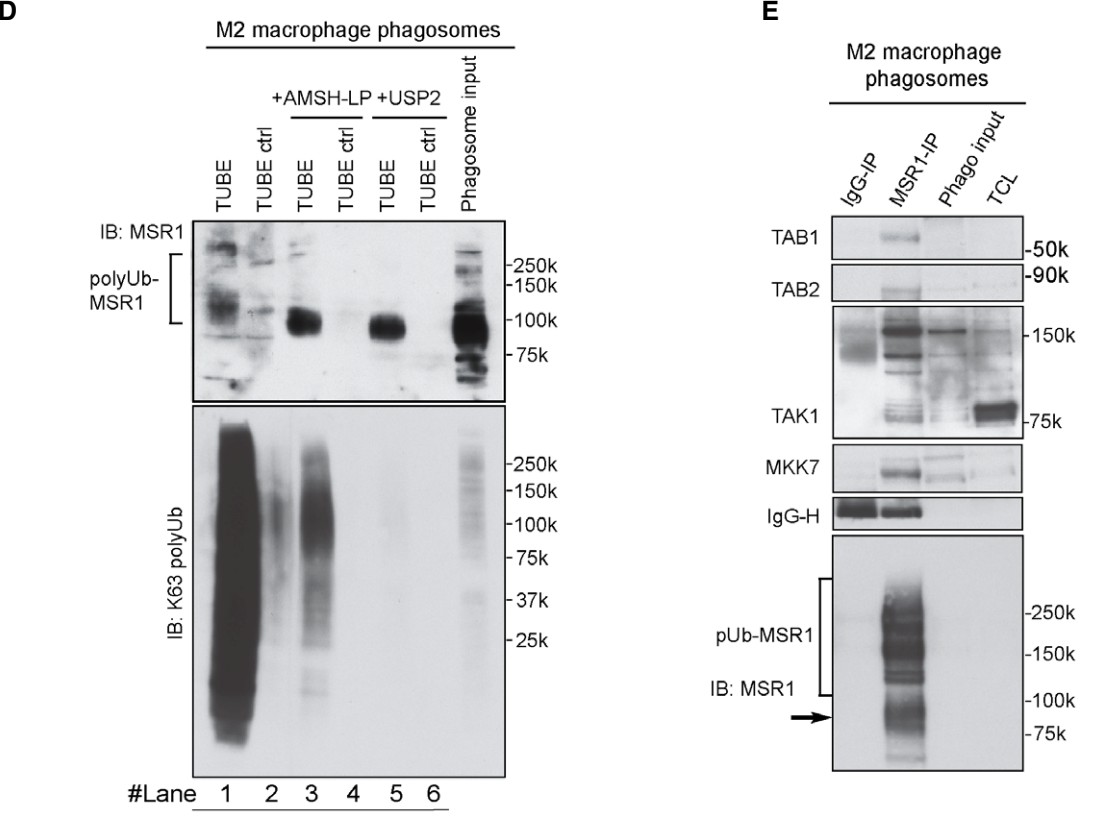

**Figure 4. Phagosomal MSR1 is K63-polyubiquitylated and interacts with Tab1/Tab2/Tak1/MKK7.**

A Workflow for TUBE pull-down of K63-polyubiquitylated proteins from phagosomal extracts.

B Selected ubiquitylated phagosomal proteins identified by TUBE-MS approach.

C Sequences alignment of N-terminal region of murine and human MSR1 shows high-sequence identity and conserved ubiquitylated lysine. The arrow points out the ubiquitylated lysine residue.

D Tab2-TUBE pull-downs from phagosome extracts of M2 (IL-4) macrophages treated with the K63-specific deubiquitylase (DUB) AMSH-LP or the unspecific DUB USP2.

E MSR1 immunoprecipitation from M2 (IL-4) macrophage phagosomes shows that TAB1, TAB2, TAK1 and MKK7 bind to polyubiquitylated MSR1. TAK1 shows a specific pattern of post-translational modifications indicative of its activation.

Data information: (D) and (E) are representative of two independent experiments.

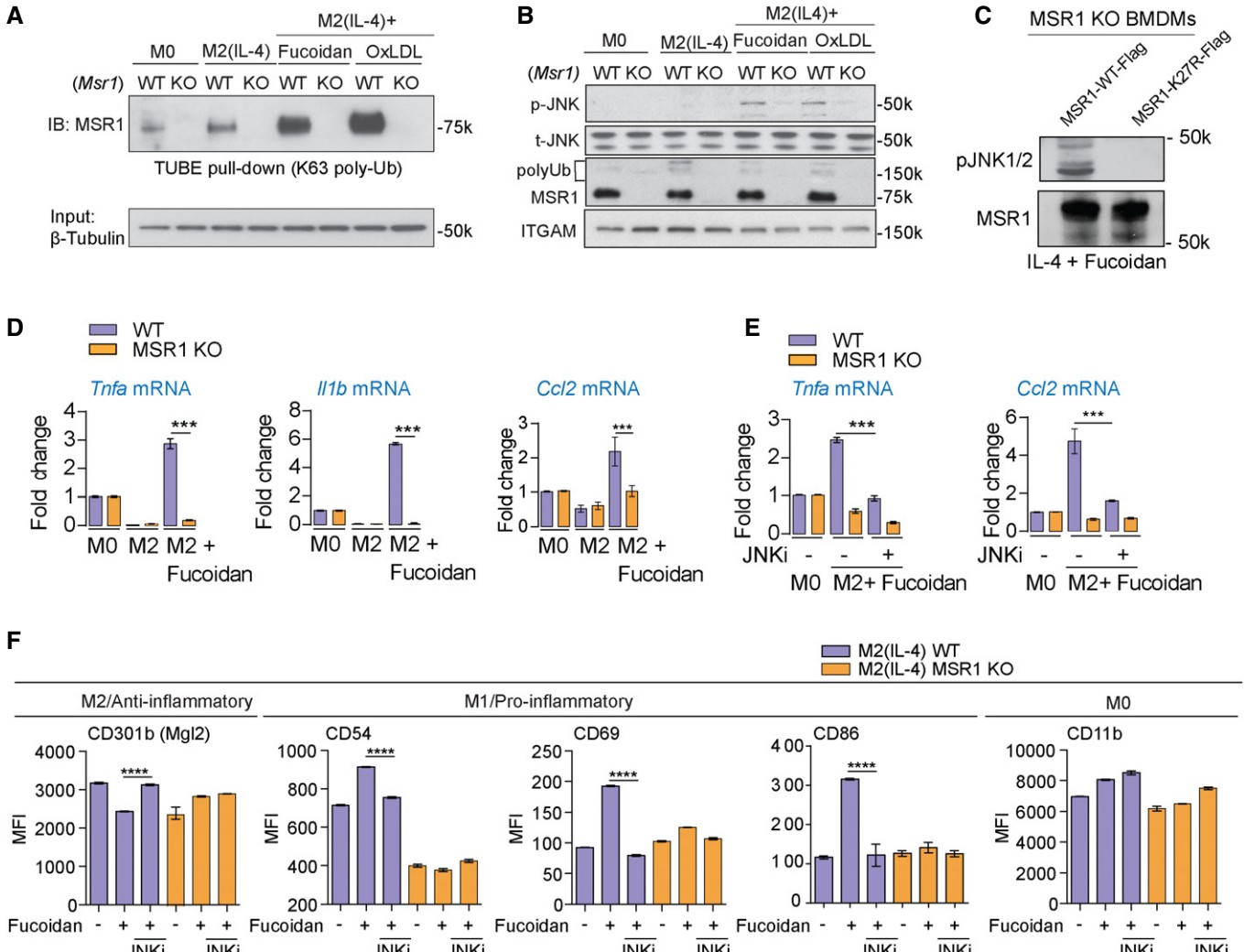

**Figure 5. MSR1 triggering activates JNK, which induces a pro-inflammatory stimulus.**

A   Immunoblot against MSR1 of TUBE pull-downs of MSR1 WT and KO BMDMs shows increasing amounts of ubiquitylated MSR1 upon alternative activation and further increases upon MSR1 ligation with fucoidan or oxLDL.

B   $Msr1^{+/+}$ (WT) and $Msr1^{-/-}$ (KO) M0 macrophages and M2 (IL-4) macrophages untreated or stimulated with the MSR1 ligands fucoidan or oxLDL (50 µg/ml, 30 min) were analysed for the phosphorylated and the total forms of JNK1/2 and MSR1. ITGAM (CD11b) serves as a loading control. Both fucoidan and oxLDL activate JNK in a MSR1-dependent manner.

C   IL4-activated MSR1 knock-out BMDMs were transfected with WT or K27R MSR1 and treated with 50 µg/ml fucoidan for 1 h. Mutation of the ubiquitylation site K27 abolishes MSR1 signalling.

D   qPCR data of Tnfa, Il1b and Ccl2 mRNA levels in WT and MSR1 KO M0 and M2 (IL-4) BMDMs show an MSR1-dependent increase in pro-inflammatory cytokines in response to MSR1 ligation by fucoidan.

E   Inhibition of JNK by JNK-IN8 reduces expression of Tnfa and Ccl2 upon MSR1 ligation, showing that it is JNK-dependent.

F   Flow cytometry analysis of cell surface markers in WT and $MSR1^{-/-}$ M0 macrophages and M2 (IL-4) macrophages untreated or stimulated with the MSR1 ligands fucoidan (50 µg/ml, 24 h). Data show MSR1-dependent increase in the early activation markers CD54, CD69 and CD86 and a decrease in the M2 marker CD301b/Mgl2. CD11b serves as a control.

Data information: Error bars represent SEM. ***$P < 0.0001$; ****$P < 10^{-5}$ (Student's t-test). (A, B) Data are representative of three independent experiments. (D, E, F) Three biological replicates.

S6B and C). This process was dependent on specific MSR1 ligands as stimulation with Toll-like receptor 4 (TLR4)-activating lipopolysaccharide (LPS) and IFN-γ did not show any differences between WT and MSR1 KO macrophages (Appendix Fig S6A) suggesting the involvement of the MSR1-JNK pathway in phenotypic switch. Finally, by blotting for the NF-kappa-B inhibitor alpha (IκBα) we could show that triggering MSR1 was not activating the NF-κB pathway (Appendix Fig S8). These data show that triggering of MSR1 in alternatively activated macrophages leads to the enhanced activation of the JNK signalling pathway, which induces a pro-inflammatory phenotype switch in these macrophages.

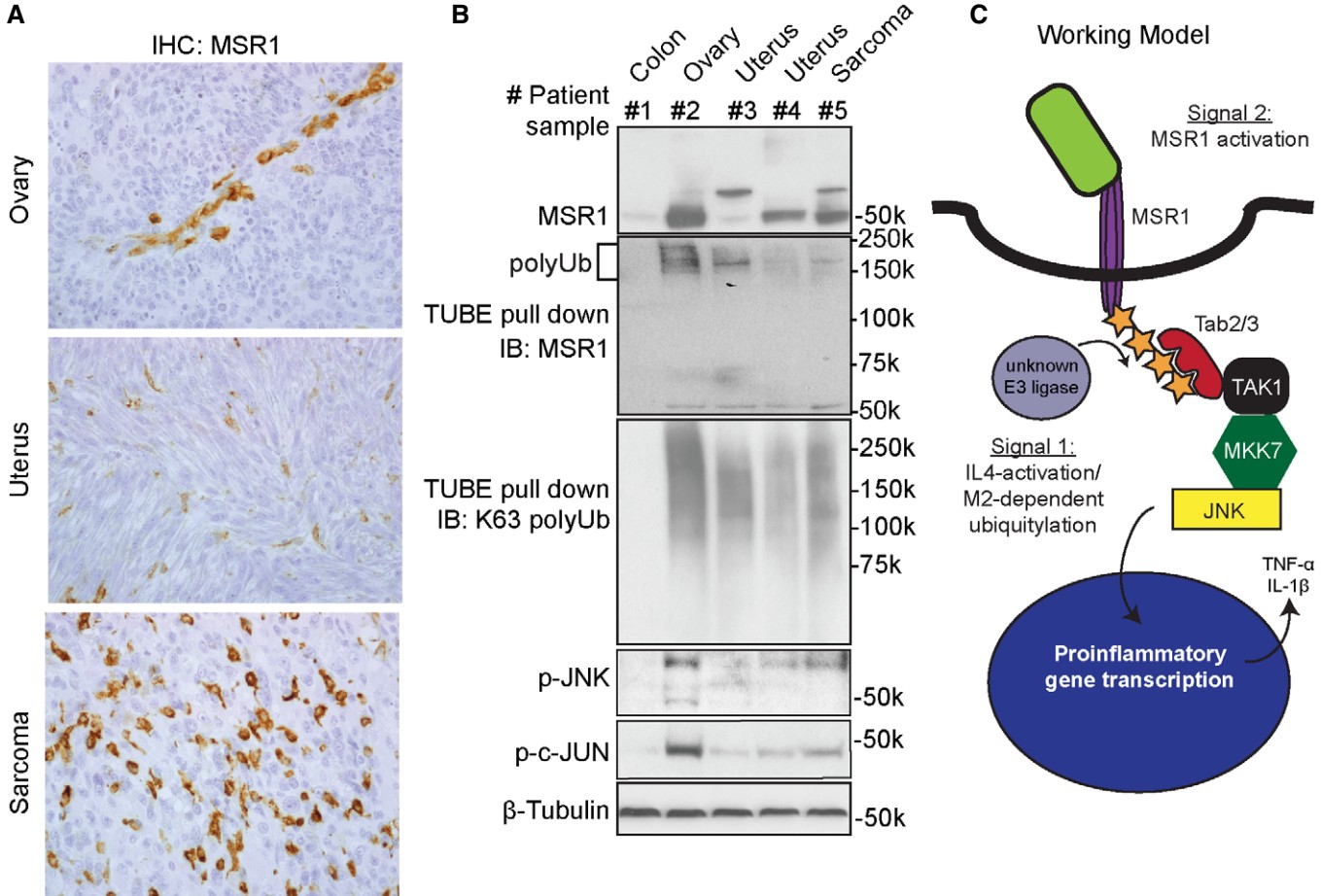

**Figure 6.  MSR1 ubiquitylation and JNK activation correlate in human tumour-associated macrophages.**

A  Immunohistochemistry analysis of MSR1 in patient ovarian, uterus and sarcoma cancers showing tumour-associated macrophages.
B  Pull-downs of K63-polyubiquitin chains and IB analysis of five human primary cancers show a correlation between the amount of polyubiquitylated MSR1 and JNK activation in an ovarian tumour.
C  Working model: MSR1 is activated by ligation through many different substrates, including apoptotic cells, fucoidan or oxidized LDL (Signal 1). However, only when the macrophage is IL-4-activated (Signal 2) becomes MSR1 ubiquitylated by an unknown E2/E3 ligase. This ubiquitylation recruits Tab2/3, Tak1, Mkk7 and finally JNK, thereby allowing MSR1 to signal directly through the JNK signalling pathway which induces pro-inflammatory gene transcription.

### MSR1 is polyubiquitylated in human tumour-associated macrophages

Next, we wanted to test whether MSR1 ubiquitylation was also present in settings of human disease. MSR1 has been implicated in tumour development and progression (Komohara *et al*, 2009; Chanmee *et al*, 2014), and tumour-associated macrophages (TAMs) have been shown to resemble M2 macrophage phenotype with MSR1 protein expression (Sica *et al*, 2007). However, the role of MSR1 in TAMs is not fully understood. We characterized five human patient samples with different types of cancers for the presence of MSR1. These tissue samples showed a high variability in the number of TAMs that stained positively for MSR1 using immunohistochemistry (Fig 6A). Consistent with the human proteome atlas data (http://www.proteinatlas.org/ENSG00000038945-MSR1/cancer), we found particularly high expression of MSR1 in patient ovarian cancer. Interestingly, the TAMs of the patient with ovarian cancer also showed increased levels of ubiquitylated MSR1 as well as enhanced

phosphorylation of JNK and its substrate c-Jun (Fig 6B). This suggests that JNK signalling downstream of polyubiquitylated MSR1 is present in TAMs of human cancers and could potentially be involved in tumour promotion.

## Discussion

Macrophages are highly diverse and plastic immune cells that can polarize in response to environmental cues into many different phenotypes. Because of their important and diverse functions in regulating immune responses and metabolism, dysregulated macrophage polarization is frequently associated with disease (Schultze *et al*, 2015). Here, we expand our understanding of macrophage phenotype switching by showing on the molecular level that engagement of MSR1 in IL-4-activated macrophages results in the activation of the JNK signalling pathway, thereby inducing a shift from an anti-inflammatory to a pro-inflammatory phenotype.

Recent data imply that distinct pathways regulate uptake kinetics of different particles as well as phagosome functions in macrophages, and these are further controlled by macrophage activation. It was demonstrated that both the phagocytic receptors (Dill *et al*, 2015; Balce *et al*, 2016) and pro-inflammatory (Yates *et al*, 2007; Trost *et al*, 2009; Ghigo *et al*, 2010) and anti-inflammatory activation (Varin *et al*, 2010; Balce *et al*, 2011) of macrophages affect phagosome functions and these are regulated by signalling pathways such as kinases (Hartlova *et al*, 2018) and E3 ligases (O. Bilkei-Gorzo, T. Heunis, A. Härtlova, M. Trost, *unpublished data*).

A striking finding from our study is the identification of increased ubiquitylation on the phagosome. Ubiquitylation in the endo-lysosomal system is generally thought to be important for lysosomal degradation of membrane proteins (Piper *et al*, 2014). Our data indicate that K63 polyubiquitylation on phagosomal proteins is also used as a scaffold for the recruitment of signalling complexes, in particular the kinase complex TAK1/MKK7/JNK to MSR1 via specific polyubiquitylation on a conserved lysine K27. Remarkably, we could show that only MKK7 and not MKK4 is recruited to this signalling complex in IL-4-activated macrophages, showing a different role for the two MAP kinase kinases in activating JNK.

It is noteworthy that MSR1 is in almost equal abundance in resting, alternatively and classically activated macrophages, but its abundance increases on the phagosome of activated macrophages. This indicates that macrophage activation (as signal 1) induces translocation of MSR1 and activation/transcription/translocation of an E3 ligase. This unknown E3 ligase then ubiquitylates the receptor after MSR1 ligation (signal 2), thereby enabling pro-inflammatory signalling through the TAK1/MKK7/JNK signalling complex (Fig 6C). During inflammatory conditions, when TLR or IL1R is activated, this signalling probably does not add substantially to the macrophage phenotype as TLR activation is much stronger a pro-inflammatory signal. However, in M2 or tissue-resident macrophages with a similar M2-like phenotype, prolonged MSR1 ligation could lead to a pro-inflammatory switch of these macrophage subsets via JNK. Interestingly, recent data in Drosophila showed that apoptotic corpses prime macrophages for detection of tissue damage and that this priming and subsequent recruitment to wounds were dependent on JNK (Weavers *et al*, 2016). Moreover, it has been shown before that JNK plays an important role in obesity-induced inflammation (Han *et al*, 2013). This suggests that JNK activation induced by uptake apoptotic bodies or lipids through MSR1 could regulate various responses.

MSR1 has in recent years been established as a good marker for TAMs (Allavena & Mantovani, 2012) which resemble rather a M2 alternatively activated phenotype and have been associated with tumour promotion (Sica *et al*, 2007). While previously it was shown that lack of MSR1 delayed the growth of EL4 lymphoma in mice by increased pro-inflammatory responses to necrotic cells (Komohara *et al*, 2009), our data revealed an increased expression of K63-polyubiquitylated MSR1 in a human ovarian cancer. This coincided with increased activation of JNK pro-inflammatory signalling pathway, suggesting that the MSR1-JNK signalling pathway is activated in the progression of cancer. Interestingly, MSR1 has been previously shown to promote tumour progression and metastasis in ovarian and pancreatic cancer mouse models (Neyen *et al*, 2013), suggesting that MSR1 and downstream signalling may be a potential drug target in the prevention of cancer metastasis progression.

# Materials and Methods

## Antibodies

The following antibodies were purchased from Cell Signalling Technology: pSTAT1 (#8826), Arginase-1 (#9819), Rab7 (#9367), Rab5 (#2143), BIP (#3183), EEA1 (#2411), IκBα (#9242), LAMP1 (#3243), Na$^+$/K$^+$ ATPase 1 (#3010), Lamin A/C (#2032), TAK1 (#4505), JNK (#9258), p-JNK (#9251), p-c-Jun (#9165) and Vimentin (#5741). Antibodies purchased from Abcam were as follows: pSTAT6 (ab54461), CD36 (ab133625), CD14 (ab182032) and MSR1 (ab151707, ab79940). Antibodies against K63-specific ubiquitin (#05-1308) and ITGAM (PAB12135) were from Millipore and Abnova, respectively. Sheep antibodies against MSR1, TAK1, TAB1, TAB2, MKK7 and MKK4, and rabbit IgG were generated by the Antibody Production Team of the Division of Signal Transduction Therapy (DSTT), Medical Research Council Protein Phosphorylation and Ubiquitylation Unit, University of Dundee, United Kingdom. The antibody used for MSR1 immunohistochemistry was clone SRA-E5 (from Abnova, #MAB1710). Commercial antibodies were used according to the manufacturer instructions. DSTT-made antibodies were used at 2 μg/ml in TBS-T containing 5% non-fat-dried milk. Recombinant proteins, plasmids and antibodies generated for the present study are available to request on our reagents website (https://mrcppureagents.dundee.ac.uk/).

## Culturing and activation of bone marrow-derived macrophages

Bone marrow cells were collected from femurs and tibiae of 6- to 8-week-old C57BL/6 wild-type (WT) or MSR1/SR-A knock-out mice (kindly provided by Siamon Gordon). The cells were treated with red blood cell lysis buffer (155 mM NH$_4$Cl, 12 mM NaHCO$_3$, 0.1 mM EDTA) and plated on tissue culture plastic (Corning Incorporated) for 3 days in DMEM (Gibco) containing 10% FBS, 2 mM glutamine, 100 units/ml penicillin–streptomycin (Gibco) and 20% L929 conditioned supplement. At day 3, the cells in supernatant were transferred to untreated 10-cm Petri dishes (BD Biosciences) for 7 days for the differentiation into bone marrow-derived macrophages (BMDMs). Then, BMDMs were treated by either IL-4 (20 ng/ml, BD Pharmingen) for 48 h to get M2 (IL-4) macrophages (Appendix Fig S1).

## Phagosome isolation

For proteomic analysis, phagosomes were isolated from BMDMs according to previous methods (Desjardins *et al*, 1994; Trost *et al*, 2009). Latex beads of 0.8 μm (Estapor/Merck, Fontenay Sous Bois, France) were diluted 1:50 in complete DMEM media and incubated with treated and untreated BMDMs for 30 min at 37°C, 5% CO$_2$. Cells were then harvested on ice, washed in cold PBS, and phagosomes were isolated as described in previous works (Trost *et al*, 2009; Guo *et al*, 2015). Enriched protein extracts contained < 5% contamination from other cellular organelles as estimated from immunoblotting experiments (Appendix Fig S2A).

For pull-down assays, phagosomes were isolated using 1-μm magnetic beads (Estapor/Merck). Magnetic beads were diluted 1:300 in complete DMEM media and incubated with BMDMs for 30 min.

## Phagosome functional assays

Fluorogenic assays for phagosomal proteolysis, acidification and lipolysis were adapted from the method from the Russell laboratory (Yates *et al*, 2005; VanderVen *et al*, 2010; Podinovskaia *et al*, 2013). For proteolysis and acidification, BMDMs were plated onto 96-well plates at $1 \times 10^5$ cells per ml 24 h prior to the experiment. DQ red BSA (Life Technologies)-coupled or BCECF (Life Technologies)-coupled carboxylated silica beads (3 μm, Kisker Biotech) were diluted 1:200 in binding buffer (1 mM $CaCl_2$, 2.7 mM KCl, 0.5 mM $MgCl_2$, 5 mM dextrose, 10 mM hydroxyethyl piperazine ethane sulfonate (HEPES) and 5% FBS in PBS pH 7.2) and incubated with BMDMs for 3 min at room temperature. Beads were replaced with warm binding buffer, and real-time fluorescence was measured at 37°C using a SpectraMax Gemini EM Fluorescence Microplate Reader (Molecular Devices), set as maximal readings per well to allow reading time intervals of 2 min. Plots were generated from the ratios of signal/control fluorescence. For lipolysis, Nucleosil 120-3 C18 reverse-phase HPLC 3 μm silica matrix was coated with a mixture of neutral lipids containing a fluorogenic substrate specific to detect the lipase activity and a calibration fluorogenic dye. Substrate-labelled beads were incubated with BMDMs on coverslips for 3 min at room temperature. The coverslips were then washed and loaded into quartz cuvettes in binding buffer. Fluorescent intensities were recorded in real time at 37°C with a thermostat-regulated QMSE4 spectrofluorometer (Photon Technologies International, Lawrenceville, NJ, USA) with excitation/emission 342/400 nm for reporter dye and 555/610 nm for calibration dye.

## Phagocytosis assays

Phagocytosis of beads was performed by incubating Alexa Flour 488 BSA-coated silica beads at 1:1,000 dilution with BMDMs in 96-well plates for 10 min at 37°C. Beads were replaced with 100 μl trypan blue to quench the fluorescence of non-internalized particles. After aspirating trypan blue, the fluorescence was measured in a SpectraMax Gemini EM Fluorescence Microplate Reader, set at excitation/emission wavelengths 495/519 nm.

## Uptake of apoptotic/necrotic cells

Uptake of apoptotic or necrotic cells was performed as follows: GFP-expressing RAW264.7 cells were induced apoptosis by 50 μM cycloheximide for 24 h or necrosis by repeated *freeze/thaw* cycles. Both apoptotic and necrotic cells were vigorously washed in PBS and then added to BMDMs for 6 h at 37°C, in a phagocyte to target ratio of approximately 1:5. The percentage of BMDMs that had interacted with apoptotic cells was quantified by FACS analysis of AF488-positive cells. A minimum of 50,000 events within the macrophage gate was acquired.

## Sample preparation and mass spectrometry analysis

Phagosome proteins were extracted using 1% sodium 3-[(2-methyl-2-undecyl-1,3-dioxolan-4-yl)methoxy]-1-propanesulfonate (commercially available as RapiGest, Waters) in 50 mM pH 8.0 Tris, reduced with 1 mM tris(2-carboxyethyl)phosphine (TCEP), and alkylated by 5 mM iodoacetamide (30 min, room temperature, in the dark) (Sigma) and then quenched by 10 mM DTT. Protein concentrations were determined using EZQ protein quantitation kit (Life Technologies). Samples were then diluted in 50 mM Tris containing 5 mM calcium chloride to a final concentration of 0.1% RapiGest and were digested by Trypsin Gold (Promega). RapiGest was removed by adding trifluoroacetic acid (TFA) to 1%, shaking at 37°C for 1 h and centrifugation at $14,000 \times g$ for 30 min. Peptides were desalted by solid-phase extraction using Microspin C-18 (Nest Group), lyophilized and labelled using mTRAQ labelling kit (Δ0 and Δ8 Da; AB Sciex) for phagosomal samples.

Mass spectrometric analyses were conducted similarly as previously described (Dill *et al*, 2015; Guo *et al*, 2015). In detail, biological triplicates or quadruplicates of mixes of 1 μg of light-labelled and 1 μg of heavy-labelled samples were analysed on an Orbitrap Velos Pro mass spectrometer coupled to an Ultimate 3000 UHPLC system with a 50 cm Acclaim PepMap 100 or EasySpray analytical column (75 μm ID, 3 μm C18) in conjunction with a PepMap trapping column (100 μm × 2 cm, 5 μm C18; Thermo Fisher Scientific). Acquisition settings were as follows: lock mass of 445.120024, MS1 with 60,000 resolution, top 20 CID MS/MS using Rapid Scan, monoisotopic precursor selection, unassigned charge states and $z = 1$ rejected, dynamic exclusion of 60 s with repeat count 1. Six-hour linear gradients were performed from 3% solvent B to 35% solvent B (solvent A: 0.1% formic acid, solvent B: 80% acetonitrile 0.08% formic acid) with a 30-min washing and re-equilibration step.

## Proteome quantification and bioinformatics analysis

The phagosome proteomic dataset of IL-4-activated macrophages was extracted from a combined analysis of phagosomes from five different activations states (IL-10, IL-13, IFN-γ and IFN-γ + IL-4; PRIDE identifier PXD004520). Only proteins identified and quantified in the IL-4-treated dataset were extracted from this. Identification and quantification were performed using MaxQuant v1.3.0.5 (Cox & Mann, 2008) with variable modifications oxidation (M), acetyl (protein N-term), deamidation (NQ), two multiplicities with mTRAQ lysine/N-terminal (Δ0 and Δ8) (for phagosome samples) or label-free and Gly-Gly (K) (for TUBE samples), maximum five modifications per peptide and two missed cleavages. Spectra were matched to a UniProt-Trembl *Mus musculus* database (50,543 entries, downloaded 18 October 2012) plus common contaminants. A reverse database was used for false peptide discovery. Mass accuracy was set to 10 ppm for precursor ions and 0.5 Da for ion trap MS/MS data. Identifications were filtered at a 1% false-discovery rate (FDR) at the protein and peptide level, accepting a minimum peptide length of 7. Quantification used only razor and unique peptides, and required a minimum ratio count of 2. "Re-quantify" and "match between runs" were enabled. Normalized ratios were extracted for each protein/condition and used for downstream analyses.

Statistical analyses were performed in Perseus (v1.3.0.4). *t*-Test-based statistics were applied on normalized and logarithmized protein ratios to extract the significantly regulated proteins. Hierarchical clustering was performed in Perseus on logarithmized ratios of significant proteins using correlation distances and average linkage to generate the heatmap.

## GO term and network analyses

The H/L log fold changes for all quantifiable proteins in each condition (in replicates) were tested against the null hypothesis that the mean log fold change was zero. We used a one-sample $t$-test with shrinkage variance of Opgen-Rhein & Strimmer (Opgen-Rhein & Strimmer, 2007). Each protein was annotated with GO terms from Mouse Genome Informatics Database (downloaded on 5/11/2014). Log fold change of each GO term was calculated as the mean of log fold changes of all proteins annotated with this GO term. The significance of this mean, against the null hypothesis that the mean is zero (i.e. there is no discernible fold change in the GO-term proteins), was found using a bootstrap technique. A protein sample of the same size as the GO-term group was randomly selected (without replacement) from the pool of all quantifiable proteins and its mean log fold change found. The sampling process was repeated 100,000 times, and the significance $P$-value was determined as the percentile of bootstraps where the absolute log fold change was greater than in the GO-term group. Proteins annotated with given GO terms are presented in "volcano plots", showing, for each protein, the mean log fold change of replicates versus the $P$-value. The error bars represent the shrinkage standard error.

Protein networks were extracted from STRING database v10 (Szklarczyk et al, 2015) using only "experiment" and "text-mining" data. Graphs were generated using Cytoscape v3.3 (Shannon et al, 2003).

## Immunoblot analysis

Cells or phagosomes were lysed directly in 2× Laemmli buffer, separated on 4-12% Nu-PAGE gels (Invitrogen) and immunoblotted onto PVDF membranes (Amersham). Membranes were blocked for 1 h in 5% (w/v) skim milk in TBS containing 0.1% (v/v) Tween 20 and subsequently incubated with different primary antibodies overnight. After incubation with HRP-labelled secondary antibodies, proteins were detected using ECL and X-ray films. Immunoblots were quantified in ImageJ software. For activation of MSR1, WT and MSR1 KO BMDMs were either stimulated with fucoidan (50 μg/ml, Sigma-Aldrich) for 1 h alone or with IL-4 (20 ng/ml, RD) for 24 h followed by fucoidan 50 μg/ml, Sigma-Aldrich) treatment for 1 h.

## Transfection of WT MSR1 and MSR1 K27R into BMDMs

MSR1 KO BMDMs were transfected either with 1 μg of pCMV WT-MSR1-FLAG or with pCMV MSR1-(K27R)-FLAG using Fugene (Promega) transfection agent for 24 h. Following day, cells were washed and stimulated with IL-4 (20 ng/ml, RD) for 24 h, followed by fucoidan (50 μg/ml, Sigma-Aldrich) treatment for 1 h.

## Immunofluorescence

Resting and alternative-activated BMDMs were seeded at $1 \times 10^5$/ml on glass coverslips. Silica beads (3 μm, Kisker Biotech) were phagocytosed for 30 min by using a dilution of 1:1,000 in cell culture media. Cells were subsequently fixed in 4% paraformaldehyde (Affymetrix) and permeabilized by incubating for 5 min with PBS containing 0.02% NP-40. Rabbit anti-Rab7a antibody was used at dilution of 1:300 to indicate the phagosomes. To visualize the phagosomal location of interested proteins, sheep anti-TAK1/TAB1/TAB2/MKK7 antibodies (DSTT) were used at 4 μg/ml. For inhibition of Ubc13, BMDMs were stimulated with IL-4 (20 ng/ml) either alone or in the presence of Ubc13 inhibitor NSC697923 (1 μM, DSTT) for 24 h. Cells were imaged in a Zeiss LSM 700 confocal microscope using a ×100 Plan Apochromat objective (NA 1.46) and an optical section thickness of 0.7 μm. For quantitation, all laser, pin-hole and gain, etc., settings kept the same for all images. Fields of cells were selected at random using only the DAPI-stained channel. Optical sections were taken through the centre of the cell including the beads and 10 fields collected per coverslip. A region of interest (ROI) was drawn around each bead-containing phagosome, the Rab7a ring associated with the bead was included. The green intensity was collected for the same ROI. The integrated sum of the red and green intensities in each ROI was collected and expressed as a ratio. At least 25 individual phagosomes were analysed for each protein target. DAPI, Green-Alexa 488 (target antigen), Red-Alexa 594 (Rab7a) and DIC channels were all collected, and images were quantified using the Volocity programme (PerkinElmer).

## TUBE pull-down

Phagosomes were isolated from M2 (IL-4) macrophages using magnetic beads. Phagosomal proteins were solubilized in cell lysis buffer (50 mM Tris–HCl pH 7.5, 1 mM EGTA, 1 mM EDTA, 1% Triton X100, 0.27 M sucrose, 0.2 mM PMSF, 1 mM benzamidine), plus 1% SDS and inhibitors of proteases, phosphatases and deubiquitylases were added freshly. Cell lysates were clarified by centrifugation at $14,000 \times g$ for 30 min at 4°C. The supernatants were collected, and their protein concentrations were determined by EZQ protein quantitation kit. For each pull-down, 500 μg of phagosome lysate was diluted in cell lysis buffer to make a final concentration of 0.1% SDS, and then incubated with Npl4 Zinc Finger (NZF) domains of TAB2 (TAB2 [644–692]) coupled beads, which were previously described (Emmerich et al, 2013). Ubiquitin-binding-defective mutant TAB2 (T674A/F675A) beads were included as negative control. After overnight incubation at 4°C, the beads were washed three times with 50 mM Tris–HCl, pH 8.0, 0.5 M NaCl, 1% Triton X-100 and eight times with 50 mM Tris–HCl, pH 8.0. Captured proteins were then eluted by incubation for 15 min with 50 mM Tris–HCl, pH 8.0, 1% RapiGest and 5 mM TCEP and centrifugation at $1,000 \times g$ for 5 min.

## Deubiquitylation assay

The polyubiquitylated proteins captured by Halo-TAB2 beads were washed twice in reaction buffer (50 mM Tris pH 7.5, 50 mM NaCl, 2 mM DTT). The beads were then incubated with or without AMSH-LP (5 μM) or USP2 (1 μM) in 30-μl reaction buffer at 30°C for 1 h. The reaction was quenched by denaturation in 1% LDS. Eluted proteins were separated on SDS–PAGE and immunoblotted with anti-MSR1 or anti-K63-pUb chain antibodies.

## MSR1 co-immunoprecipitation from phagosome extracts

Rabbit anti-MSR1 antibody and rabbit IgG were coupled to protein A-Sepharose (Amersham Biosciences) by incubation in PBS for 5 h at 4°C. Then, antibodies were cross-linked to the Sepharose by

incubating with 20 mM dimethyl pimelimidate dihydrochloride (DMP, Sigma) in 0.1 M sodium tetraborate decahydrate (Sigma) pH 9.3 for 30 min at room temperature. Excess antibody was removed by washing the Sepharose in 50 mM glycine (Sigma) pH 2.5. The resin was washed extensively with 0.2 M Tris–HCl pH 8.0. Phagosomes were isolated from M2 (IL-4) macrophages using magnetic beads. Phagosomal lysate of 500 μg was incubated with antibody cross-linked Sepharose in cell lysis buffer for 5 h at 4°C. The beads were washed three times with 1 ml of lysis buffer. Immunoprecipitated proteins were eluted by denaturation in 1% LDS and subjected to immunoblotting.

### Preparation of oxLDL

Human low-density lipoprotein (Millipore) was oxidized by incubating at 1 mg/ml with 5 μM $CuSO_4$ (Sigma) in PBS for 18 h at 37°C. The oxidized low-density lipoprotein (oxLDL) was dialysed in a dialysis cassette (Pierce) against PBS to remove $CuSO_4$ and then added to BMDMs at the desired concentration.

### Quantitative real-time PCR

RNA was isolated from cells using RNeasy kit from Qiagen. Total RNA (500–1,000 ng) was used to synthesize cDNA with the QuantiTect Reverse Transcription Kit (Qiagen). Quantitative real-time PCR analysis was performed using the iTaq™ Universal SYBR® Green Supermix and analysed using the CFX384 Touch™ Real-Time PCR Detection System (Bio-Rad). The results were normalized to *Gapdh* and expressed as fold change relative to RNA samples from control or mock-treated cells using the comparative CT method ($\Delta\Delta C_T$). The following validated QuantiTect primer assays (Qiagen) were used: *Gapdh*, *Il1b* and *Tnfa*.

Ccl2 was analysed using the following primer sequences: forward primer: ATTCTGTGACCATCCCCTCAT; reverse primer: TGTATGT GCCTCTGAACCCAC.

### Flow cytometry analysis

M0 and M2 (IL-4) were incubated with fucoidan from *Fucus vesiculosus* (50 μg/ml; Sigma-Aldrich) and oxLDL (50 μg/ml) for 24 h, after which cells were collected at $1 \times 10^6$ cells per tube into polystyrene tubes (Corning Incorporated). Cells in each tube were washed twice in PBS/FBS (1% FBS in PBS, pH 7.2) by centrifugation at $1,200 \times g$ for 3 min and resuspended in 100-μl blocking buffer (1–50 dilution of CD16/CD32 in PBS/FBS) to block Fc receptors. Following 15-min incubation in ice, cells were washed once in PBS/FBS and then incubated with 100 μl PBS/FBS containing 1–100 dilution of phycoerythrin (PE)-conjugated CD54, CD69, CD86, CD301b or CD36 antibodies (eBioscience) for 30 min on ice, respectively. PE-CD11b (eBioscience) was mixed with each antibody at 1–200 dilution as control. After washing for three times in PBS/FBS, cells were analysed by flow cytometry.

### JNK inhibitor treatment

BMDMs were pre-treated with 0.3 μM JNK7 or JNK8 inhibitor (Zhang *et al*, 2012) 1 h before stimulation with 50 μM of fucoidan for the indicated time (6 h—RT–PCR analysis, 24 h—FACS analysis).

### Immunohistochemistry

Three-micrometre tissue sections from selected paraffin blocks of primary human tumours were prepared. Slides were incubated overnight at 56°C, deparaffinized in xylene for 20 min, rehydrated through a graded ethanol series and washed with PBS. Immunohistochemistry was performed on a Ventana Benchmark XT automatic immunostaining device (Roche). A heat-induced epitope retrieval step was performed in Ventana CC1 solution for 60 min. Primary antibodies were incubated for 40, 60 and 120 min, respectively. An Ultravision detection system was used.

### Statistical analysis

Statistical analysis was performed using GraphPad Prism software. Definition of statistical analysis and *post hoc* tests used can be found in figure legends. The statistical significance of data is denoted on graphs by asterisks (*) where $*P < 0.05$, $**P < 0.01$, $***P < 0.001$ or ns = not significant.

# Data availability

Mass spectrometric raw data are available through the PRIDE repository (https://www.ebi.ac.uk/pride/archive/) and have been assigned the identifiers PXD010478 (TUBE pull-down data) and PXD004520 (phagosome proteomic data).

**Expanded View** for this article is available online.

## Acknowledgements

We would like to thank the DNA cloning, protein production, antibody production, DNA sequencing facility, tissue culture and mass spectrometry teams of the MRC Protein Phosphorylation and Ubiquitylation Unit for their support. We would like to thank Rosemary Clarke for help with flow cytometry analyses, Sir Philip Cohen for providing reagents, Siamon Gordon for kindly providing MSR1 KO mice, Natalia Shpiro for synthesis of sodium 3-[(2-methyl-2-undecyl-1,3-dioxolan-4-yl)methoxy]-1-propanesulfonate and Carol Clacher, Laura Frew and Gail Gilmour in Transgenic Services for collection of murine femurs. This work was funded by Medical Research Council UK (MC_UU_12016/5) and the pharmaceutical companies supporting the Division of Signal Transduction Therapy (DSTT) (Boehringer-Ingelheim, GlaxoSmithKline and Merck KGaA). ASH is funded by the Knut and Alice Wallenberg Foundation and the Wallenberg Centre for Molecular and Translational Medicine, University of Gothenburg, Sweden. We would like to acknowledge Dundee Imaging Facility which is supported by the "Wellcome Trust Technology Platform" award (097945/B/11/Z) and the "MRC Next Generation Optical Microscopy" award (MR/K015869/1). The School of Life Sciences Data Analysis Group and the Flow Cytometry Facility are funded by Wellcome Trust grants 097945/Z/11/Z and 081867/Z/06/Z, respectively.

## Author contributions

MGu and AH performed most experiments; AP, JC, JHL, BDD, CHE, SKP, UAW and DGR performed additional experiments; MGu, MGi and MT performed data analysis; SRYC and DGR provided intellectual input; MT, AH and MGu designed experiments; MT, AH and MGu wrote the paper with contributions of all authors.

## Conflict of interest

The authors declare that they have no conflict of interest.

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
