## [Review Process File · The EMBO Journal]

Triggering MSR1 promotes JNK-mediated inflammation in IL-4 activated macrophages

Manman Guo, Anetta Härtlova, Marek Gierliński, Alan Prescott, Josep Castellvi, Javier Hernandez Losa, Sine K. Petersen, Ulf A. Wenzel, Brian D. Dill, Christoph H. Emmerich, Santiago Ramon Y Cajal, David G. Russell and Matthias Trost

Review timeline:

Submission date:	1st Aug 2018
Editorial Decision:	24th Sep 2018
Revision received:	6th Feb 2019
Editorial Decision:	22nd Mar 2019
Accepted:	26th Mar 2019

Editor: Karin Dumstrei

Transaction Report:

1st Editorial Decision

24th Sep 2018

Thank you for submitting your manuscript to The EMBO Journal. Sorry for the delay in getting back to you with a decision, but I have now received the input from two referees and their comments are provided below.

As you can see the referees appreciate that the analysis, but also find that some further experiments are needed to fully support the key conclusions. Should you be able to address the raised concerns in full then I would like to invite you to submit a revised version. Let me know if we need to discuss any specific experiments further.

REFeree REPORTS:

Referee #1:

This study appears technically well done. Conceptually, it may provide a new insight into a particular pathway potentially capable of or contributing to switching M2s to M1 macrophages. Since this pathway was tapped through unbiased MS data, it appears that it may be important. This is further supported by some data from ovarian cancer tissues.

I nevertheless have mixed impressions of this work:

A) I find this work technically solid (with some exceptions re missing techniques) and replete with informative sets of data, albeit not necessarily coming fully together in a comprehensive story as one would hope. Regarding how these data are combined and presented it seems that there may be a

need or rather an opportunity for better organization of "why" certain experiments were done and "what" the conclusions were.

B) This reviewer has very little issues with technical aspects of this study but is frankly overwhelmed by the flow or rather lack of it and - with all due respect and apologies - non sequiturs in interpretations, which appear too many to list them all, with just a subset highlighted below. It is incumbent upon the authors to either take this into account and adjust, and then apply similar measures throughout, or to ignore altogether. Only after such a comprehensive overhaul can this be re-reviewed properly. Below is a sampling of issues (note: not a comprehensive and full coverage).

1. The abstract is OK but it also can be improved in terms of conceptual line of thought. It is a bit confusing to start with phagosomal maturation and then switch to M2/M1 reprogramming (which also seems not to be clearly stated). Please modify.
2. Fig. 2B. TAK1 was on the volcano plot in a very "gray" area of lesser standouts (by fold change and p values it is at the intersection of the cutoffs). Why focus on it then?
3. Fig. 1 A-B. The uptake of apoptotic vs necrotic cells. How does that fit with the main focus of the study? Why is in this study the charge analyzed? Will this contribute to the overall main storyline of this work? Are the differences in uptake (albeit labeled statistically significant) really that different? They are presented as different, but than the figure title says that phagocytosis is not affected.
4. Real time acidification, proteolysis, lipolysis.... This needs to be better explained in the legend so that the reader does not need to wade through methods. Needs to be clear what method was used and how were these measurements done (Fig. 1C-E), rather than "real time measurements". Also appropriate repeats and stats need to be shown along with the tracings.
5. The polyubiquitination of MSR1: Is this the only receptor whose polyubiquitination affects the outcome? The authors need to use mutants in a more coherent way to test whether this is a correlate or a driver of processes.
6. The authors state in the discussion that MSR1 is not important for inflammatory switching, as TLR and IL1R is more important. So what is this important for? This is definitely not clear.
7. Related to the above point (6): There may be something else that the authors missed to think of that's downstream of TAK1.

Etc., etc....

Referee #2:

The manuscript by Guo et al., entitled "Triggering MSR1 promotes JNK-mediated inflammation in IL-4 activated macrophages" demonstrates a phagosomal signaling pathway in M2 macrophages. The authors found that activated MSR1 caused ubiquitin induced TAK1/MKK7/JNK signaling in IL-4 activated M2 macrophages, thereby allowing macrophage phenotypic switching from the M2 (anti-inflammatory) toward the M1 (inflammatory) state.

The studies were conducted using multiple different biological approaches and provided novel scientific evidence on a molecular level, establishing the phagosomal MSR1-JNK signaling pathway in macrophages. However, I have some general concerns that should be addressed. Moreover, some additional studies are required to more fully document the presented conclusions.

Detailed comments are as follows:

1. The authors used an IL4-induced M2 polarization model in this study, but did not examine other M2 polarization conditions, for example IL13-induced M2 polarization or other types of polarization, including M2b and M2c. The authors need to conduct phagocytosis assays in these different M2 polarization conditions. These experiments will clarify whether the authors' conclusions are restricted to only IL4-mediated M2 polarization or whether the authors' conclusions are relevant to other types of M2 polarization.
2. In Figure 3A, JNK protein expression in the IL4-treated phagosomal fraction is unclear. The authors should provide more convincing data. In Figure 3B, the representative quantification data do not seem to correlate with the data presented in Figure 3A. The authors should provide the

quantification method. Why does the TCL fraction not contain MKK7 protein?

3. The authors showed that ubiquitylated MSR1 is recruited to the Tab1/Tab2/Tak1/MKK7 complex using the UBC13 inhibitor in Figure 3D. The authors should also provide immunocytochemistry data for the Tab1, Tab2, and MKK7 proteins using the UBC13 inhibitor in Figure S3.

4. In Figure S4, the authors showed MSR1 protein expression in M0 and M2 macrophages by western blot. The author should provide western blot data for MSR1 protein expression in IFN γ -treated M1 macrophages in Figure S4.

5. The authors state " MSR1 showed diminished MSR1 K63-polyubiquitylation" according to Figure 5B. However, the MSR1 ubiquitylation data are unclear. The authors should provide better data to support this conclusion. In addition, the authors should provide p-MKK7 western blot data. Regarding Figure 5B, do Fucoidan or OxLDL alone activate the JNK signaling pathway? The authors should include data for Fucoidan and OxLDL alone treatments in Figure 5A, 5B, 5C, and 5D.

6. In Figure 5C & 5D, the authors need to provide data for additional M1 marker expression. This manuscript would be much more convincing for the conclusion that MSR-1 promotes pro-inflammatory M1 markers expression.

7. The authors do not show CD86 data in Figure 5E. The authors should provide these data.

8. In Figure S5, the authors should provide quantification graphs.

9. The authors should co-stain for a macrophage marker (e.g. F4/80) and MSR1 in Figure 6A.

10. The authors' conclusion that the JNK pathway promotes M1 polarization has been reported previously and should be cited (PMID:23223452).

There are also several minor comments:

1. In line 5 of page 4, "Keizer, S.J." should be removed.
2. In line 4, page 7, "Figure 1A" should be "Figure 2A".
3. In line 15, page 7, "Figure 2D should be "Figure 2C".
4. In line 16, page 7, "Figure 2E" should be "Figure 2D, 2E"
5. In line 23, page 7, "Figure 2E should be "Figure 2C".
6. In line 24, page 11, "Figure 4E should be "Figure 4D".
7. In line 8, page 12, "Figure 4F should be "Figure 4E".
8. In Figure 1 legend, "macroophages" should be "macrophages"
9. In Figure 3 legend, "(F)" should be "(D)".
10. In Figure 4 legend, "(F)" should be "(E)".

We thank both referees for their insightful comments.

Referee #1:

This study appears technically well done. Conceptually, it may provide a new insight into a particular pathway potentially capable of or contributing to switching M2s to M1 macrophages. Since this pathway was tapped through unbiased MS data, it appears that it may be important. This is further supported by some data from ovarian cancer tissues.

I nevertheless have mixed impressions of this work:

A) I find this work technically solid (with some exceptions re missing techniques) and replete with informative sets of data, albeit not necessarily coming fully together in a comprehensive story as one would hope. Regarding how these data are combined and presented it seems that there may be a need or rather an opportunity for better organization of "why" certain experiments were done and "what" the conclusions were.

We thank the reviewer for their encouraging comments. We decided to present the data in the same order as we identified them. This makes it more like a "detective story". We decided against making up a hypothesis and telling the story the other way around, as we believe it is important for readers to see that unbiased proteomics data can result in surprising and exciting new findings.

*We took the reviewer's concerns on board and described better **why** we did certain experiments.*

B) This reviewer has very little issues with technical aspects of this study but is frankly overwhelmed by the flow or rather lack of it and - with all due respect and apologies - non sequiturs in interpretations, which appear too many to list them all, with just a subset highlighted below. It is incumbent upon the authors to either take this into account and adjust, and then apply similar measures throughout, or to ignore altogether. Only after such a comprehensive overhaul can this be re-reviewed properly. Below is a sampling of issues (note: not a comprehensive and full coverage).

1. The abstract is OK but it also can be improved in terms of conceptual line of thought. It is a bit confusing to start with phagosomal maturation and then switch to M2/M1 reprogramming (which also seems not to be clearly stated). Please modify.

We re-wrote the abstract intensively. We hope it is now satisfactory for the reviewer.

2. Fig. 2B. TAK1 was on the volcano plot in a very "gray" area of lesser standouts (by fold change and p values it is at the intersection of the cutoffs). Why focus on it then?

We focused on TAK1 and MKK7 as they were significantly changing, known to be forming a complex and as it was rather surprising to find a pro-inflammatory signalling complex increased on the phagosome of anti-inflammatory macrophages.

3. Fig. 1 A-B. The uptake of apoptotic vs necrotic cells. How does that fit with the main focus of the study? Why is in this study the charge analyzed? Will this contribute to the overall main storyline of this work? Are the differences in uptake (albeit labeled statistically significant) really that different? They are presented as different, but than the figure title says that phagocytosis is not affected.

This has been changed. We added a sentence that negatively charged particles are taken up through scavenger receptors, thereby linking the data better to later results.

4. Real time acidification, proteolysis, lipolysis.... This needs to be better explained in the legend so that the reader does not need to wade through methods. Needs to be clear what method was used and how were these measurements done (Fig. 1C-E), rather than "real time measurements". Also appropriate repeats and stats need to be shown along with the tracings.

We added more detail to the figure legend as requested. SEM traces are shown for figures 1C & D (very small as did many, many replicates). Figure 1E is a representative of 3 independent replicates.

5. The polyubiquitination of MSR1: Is this the only receptor whose polyubiquitination affects the outcome? The authors need to use mutants in a more coherent way to test whether this is a correlate or a driver of processes.

There are a few other receptors in the list of ubiquitylated proteins (Table EV2/EV3). However, MSR1 was the most enriched receptor with 129-fold increase between TAB2-TUBE and the control-mutant Tab2-TUBE. The peptide with the ubiquitylation site on K27 of MSR1 was only identified in the TUBE pull-down and not in control. Finally, it was the 3rd most intense Gly-Gly peptide after ubiquitin itself and Fc-receptor gamma, which was only 6-fold enriched and thus not further followed up.

We have now done additional experiments to address the role of MSR1 ubiquitylation. The data shows that point mutation of MSR1 at K27R abrogated the activation of JNK in response to IL-4+Fucoidan indicating that MSR1 polyubiquitylation is required for JNK signalling activation (Figure 5C).

6. The authors state in the discussion that MSR1 is not important for inflammatory switching, as TLR and IL1R is more important. So what is this important for? This is definitely not clear.

We have changed this part to make it clearer. The signalling is not as strong as acute LPS/TLR signalling, which is clearly a very strong inflammatory signal. However, the signal is strong enough to switch macrophages to an inflammatory phenotype and it will play an important role in chronic conditions. We believe that this will play an important role in conditions where MSR1 will be continuously triggered by lipids. We have started a project in which we could confirm that chronic inflammation triggered by obesity is dependent on this MSR1 signalling. Including this early data would be out of scope for this manuscript, though. We would like to invite the reviewer to watch out for another paper from our group in the future.

7. Related to the above point (6): There may be something else that the authors missed to think of that's downstream of TAK1.

We have tested the activation of NF-kB as suggested by testing for IkbA degradation (Figure EV8). Under our experimental settings, WT and MSR1 KO BMDMs stimulated either with Fucoidan alone or with IL-4 followed by Fucoidan stimulation do not exhibit the activation of NF-kB signalling pathway. It appears likely that the proinflammatory switch is only mediated through the JNK pathway.

Etc., etc....

Referee #2:

The manuscript by Guo et al., entitled "Triggering MSR1 promotes JNK-mediated inflammation in IL-4 activated macrophages" demonstrates a phagosomal signaling pathway in M2 macrophages. The authors found that activated MSR1 caused ubiquitin induced TAK1/MKK7/JNK signaling in IL-4 activated M2 macrophages, thereby allowing macrophage phenotypic switching from the M2 (anti-inflammatory) toward the M1 (inflammatory) state.

The studies were conducted using multiple different biological approaches and provided novel scientific evidence on a molecular level, establishing the phagosomal MSR1-JNK signaling pathway in macrophages.

However, I have some general concerns that should be addressed. Moreover, some additional studies are required to more fully document the presented conclusions.

Detailed comments are as follows:

1. The authors used an IL4-induced M2 polarization model in this study, but did not examine other M2 polarization conditions, for example IL13-induced M2 polarization or other types of polarization, including M2b and M2c. The authors need to conduct phagocytosis assays in these different M2 polarization conditions. These experiments will clarify whether the authors' conclusions are restricted to only IL4-mediated M2 polarization or whether the authors' conclusions are relevant to other types of M2 polarization.

We have now included additional data showing that IL13-activate macrophage behave like IL-4 activated macrophages. Both M2 type cytokines, IL-4 and IL-13, increased the uptake of apoptotic cells and negatively charged carboxylated beads, while the uptake of necrotic cells and positively charged amino beads was comparable to untreated control.

2. In Figure 3A, JNK protein expression in the IL4-treated phagosomal fraction is unclear. The authors should provide more convincing data. In Figure 3B, the representative quantification data do not seem to correlate with the data presented in Figure 3A. The authors should provide the quantification method. Why does the TCL fraction not contain MKK7 protein?

We provided a more representative Western blot from three biological replicates. This is cut out of a blot with other conditions and we have added the full blot in Figure EV4C. Please note that for one sample for a western blot of latex bead phagosomes ~21 10 cm dishes of BMDMs are required. The quantification method (imageJ) was added to the figure legend of Figure 3.

3. The authors showed that ubiquitylated MSR1 is recruited to the Tab1/Tab2/Tak1/MKK7 complex using the UBC13 inhibitor in Figure 3D. The authors should also provide immunocytochemistry data for the Tab1, Tab2, and MKK7 proteins using the UBC13 inhibitor in Figure S3.

We have now included an additional experiment. The Ubc13 inhibitor treatment, as expected, significantly decreased the enrichment of MKK7 on the phagosome of IL-4 treated macrophages.

4. In Figure S4, the authors showed MSR1 protein expression in M0 and M2 macrophages by western blot. The author should provide western blot data for MSR1 protein expression in IFN γ -treated M1 macrophages in Figure S4.

We have added mass spectrometry data from phagosome proteomics experiments of IFN-g, IL-4, IL-13 and IL-10 activated macrophages. These data show that all these activation states increase the amount of MSR1 on phagosomes to about the same levels (~1.7 to 2.4-fold).

5. The authors state " MSR1 showed diminished MSR1 K63-polyubiquitylation" according to Figure 5B. However, the MSR1 ubiquitylation data are unclear. The authors should provide better data to support this conclusion. In addition, the authors should provide p-MKK7 western blot data. Regarding Figure 5B, do Fucoïdan or OxLDL alone activate the JNK signaling pathway? The authors should include data for Fucoïdan and OxLDL alone treatments in Figure 5A, 5B, 5C, and 5D.

We think we did not make it clear enough that the blot in 5A shows a TUBE pull-down specific for K63 ubiquitylated proteins. Thus, the pulled-down MSR1 for which we blot is polyubiquitylated. We have changed the figure accordingly.

We have now included new data. Under our experimental settings, WT BMDMs stimulated with Fucoidan exhibit lower activation of Mkk7 and JNK compared to WT BMDMs co-stimulated with IL-4 followed by Fucoidan stimulation. As briefly noted in the discussion, we hypothesized that IL-4 induces specific E3 ligase activation that is involved in the regulation of signalling protein complex formation and promotes MKK7-JNK activation. (Figure EV8).

6. In Figure 5C & 5D, the authors need to provide data for additional M1 marker expression. This manuscript would be much more convincing for the conclusion that MSR-1 promotes pro-inflammatory M1 markers expression.

We performed additional experiments and included qPCR data for Ccl2 in Figure 5D & E and flow data for CD86 in Figure 5F.

7. The authors do not show CD86 data in Figure 5E. The authors should provide these data.

We performed additional experiments and included flow data for CD86 in Figure 5F.

8. In Figure S5, the authors should provide quantification graphs.

Done. We provided the additional quantification of the FACS data in Figure EV7.

9. The authors should co-stain for a macrophage marker (e.g. F4/80) and MSR1 in Figure 6A.

Unfortunately, these cancer slides are not available for us anymore and thus, we cannot do as requested. But considering that MSR1 (CD204) is considered a solid marker for tumour associated macrophages (Li et al, Lung Cancer. 2018; Sun et al, Biomed Res Int, 2018, Kawachi et al, Cancer Sci, 2018; Miyasato et al, Cancer Sci, 2017 and many more), we are convinced that the stained cells are indeed macrophages.

10. The authors' conclusion that the JNK pathway promotes M1 polarization has been reported previously and should be cited (PMID:23223452).

We apologise for not including this reference, which slipped out from a previous version. We included it.

There are also several minor comments:

1. In line 5 of page 4, "Keizer, S.J." should be removed.

Done.

2. In line 4, page 7, "Figure 1A" should be "Figure 2A".

Done.

3. In line 15, page 7, "Figure 2D" should be "Figure 2C".

Done.

4. In line 16, page 7, "Figure 2E" should be "Figure 2D, 2E".

Done.

5. In line 23, page 7, "Figure 2E" should be "Figure 2C".

Done.

6. In line 24, page 11, "Figure 4E" should be "Figure 4D".

Done.

7. In line 8, page 12, "Figure 4F" should be "Figure 4E".

Done.

8. In Figure 1 legend, "macroophages" should be "macrophages".

Done.

9. In Figure 3 legend, "(F)" should be "(D)".

Done.

10. In Figure 4 legend, "(F)" should be "(E)".

Done.

2nd Editorial Decision

13th Mar 2019

Thank you for submitting your revised manuscript to The EMBO Journal. Your revised manuscript has seen by referee #2 and the comments are provided below. As you can see, the referee appreciates the introduced changes and support publication here. There are a few issues that need to be sorted out please also make sure to discuss the Han et al citation in a better manner.

REFeree REPORTS:

Referee #2:

The revised paper has been improved. The conclusion that MSR1 triggers JNK-mediated inflammation in IL4-treated macrophages is supported by the data shown. The paper provides an advance that deserves to be published. However, two remaining issues need to be addressed:

a) I saw no Figure legends in the version of the manuscript that I reviewed. This omission made it difficult to evaluate the paper. This must be corrected.

b) The authors corrected an omission in the previous version of the manuscript by including a citation to Han et al. (lines 393-395). This is an improvement. However, my impression is that this could have been done in a more forthright manner that more accurately reflects this literature. While Han et al. did study obesity-associated inflammation (noted by the current authors in their citation), Han et al. also reported the effects of compound JNK-deficiency in BMDMs under M1 and M2 polarizing conditions in vitro, which appears to be directly relevant to the current study and is surprisingly unstated.

2nd Revision - authors' response

22nd Mar 2019

The authors performed all requested editorial changes.

Corresponding Author Name: Matthias Trost

Journal Submitted to: EMBO J

Manuscript Number: EMBOJ-2018-100299R